

# Pore-scale permeability prediction for Newtonian and non-Newtonian fluids

Philipp Eichheimer[1], Marcel Thielmann[1], Anton Popov[2], Gregor J. Golabek[1], Wakana Fujita[3], Maximilian O. Kottwitz[2], and Boris J.P. Kaus[2]

[1]Bayerisches Geoinstitut, University of Bayreuth, Universitätsstrasse 30, 95447 Bayreuth, Germany
[2]Institute of Geoscience, Johannes Gutenberg University, Johann-Joachim-Becher-Weg 21, 55128 Mainz, Germany
[3]Department of Earth Science, Tohoku University, 6-3, Aramaki Aza-Aoba, Aoba-ku, Sendai 980-8578, Japan

**Correspondence:** Philipp Eichheimer (Philipp.Eichheimer@uni-bayreuth.de)

**Abstract.** The flow of fluids through porous media such as groundwater flow or magma migration are key processes in geological sciences. Flow is controlled by the permeability of the rock, thus an accurate determination and prediction of its value is of crucial importance. For this reason, permeability has been measured across different scales. As laboratory measurements exhibit a range of limitations, the numerical prediction of permeability at conditions where laboratory experiments struggle

has become an important method to complement laboratory approaches. At high resolutions, this prediction becomes computationally very expensive, which makes it crucial to develop methods that maximize accuracy. In recent years, the flow of non-Newtonian fluids through porous media has gained additional importance due to e.g., the use of nanofluids for enhanced oil recovery. Numerical methods to predict fluid flow in these cases are therefore required.

Here, we employ the open-source finite difference solver LaMEM to numerically predict the permeability of porous media

at low Reynolds numbers for both Newtonian as well as non-Newtonian fluids. We employ a stencil rescaling method to better describe the solid-fluid interface. The accuracy of the code is verified by comparing numerical solutions to analytical ones for a set of simplified model setups. Results show that stencil rescaling significantly increases the accuracy at no additional computational cost. Finally, we use our modeling framework to predict the permeability of a Fontainebleau sandstone, and demonstrate numerical convergence. Results show very good agreement with experimental estimates as well as with previous

studies. We also demonstrate the ability of the code to simulate the flow of power law fluids through porous media. As in the Newtonian case, results show good agreement with analytical solutions.

## 1 Introduction

Fluid flow within rocks is of interest for several Earth Science disciplines including petrology, hydrogeology and petroleum geoscience, as fluid flow is relevant to the understanding of magma flow, groundwater flow, and oil flow respectively (Manwart

et al., 2002). Permeability estimates can be inferred on several scales ranging from macroscale (crust) (Fehn and Cathles, 1979; Norton and Taylor Jr, 1979) over mesoscale (e.g. bore hole) (Brace, 1984) to pore scale (e.g. laboratory) (Brace, 1980). Permeability at crustal scale is of great importance as crustal scale permeability is a function of its complex microstructure, therefore an accurate prediction of permeability on the pore scale is necessary (Mostaghimi et al., 2013). Typical limitations





for laboratory measurements on pore scale are: (i) change of the sample's microstructure and therefore its physical properties through cracking and self-filtration (Zeinijahromi et al., 2016; Dikinya et al., 2008) (ii) pressure changes due to the influence of wall effects (Ferland et al., 1996) and finally (iii) difficulties to measure on irregular grain shapes and small grain sizes of the porous medium (Cui et al., 2009; Gerke et al., 2015).

At this point numerical modelling can help to compute permeabilities and understand the microstructures as well as flow patterns in three dimensional pore structures. To compute fluid flow directly within 3D pore structures it is necessary to determine the morphology of the investigated sample. This can be achieved by digital rock physics (DRP). It is a powerful tool which allows to improve the understanding of both pore scale processes and rock properties. DRP approaches use 2D or 3D microstructural images to compute fluid flows (Fredrich et al., 1993; Ferreol and Rothman, 1995; Keehm, 2003; Bosl et al.,

1998), which are obtained using modern techniques including x-ray computer tomography and magnetic resonance imaging (Dvorkin et al., 2011; Arns et al., 2001; Arns, 2004). In a first step the obtained microstructural images undergo several stages of segmentation (binarization, smoothing etc.) necessary to create a three dimensional pore space. The subsequent computation of fluid flow through the reconstructed three dimensional pore space is tackled with either Lattice- Boltzmann (Bosl et al., 1998; Pan et al., 2004; Guo and Zhao, 2002) , Finite Difference (Manwart et al., 2002; Shabro et al., 2014; Gerke et al., 2018) or

Finite Element methods (Garcia et al., 2009; Akanji and Matthai, 2010; Bird et al., 2014). The computed velocity field is then used to estimate permeability (Keehm, 2003; Saxena et al., 2017) and other physical properties (Saxena and Mavko, 2016; Knackstedt et al., 2009).

In recent years, the flow of non-Newtonian fluids has gained significant interest due to their use in a wide range of applications including geology, medicine and other industrial processes (e.g. Suleimanov et al., 2011; Johnston et al., 2004; Choi,

2009). It has been shown that nanofluids displaying a non-Newtonian rheology significantly enhance the efficiency of oil recovery (Wasan and Nikolov, 2003; Huang et al., 2013), which makes it necessary to develop numerical models that can simulate non-Newtonian flow through porous media.

In this paper we enhance the open-source finite difference solver LaMEM to model fluid flow on pore-scale with both Newtonian as well as non-Newtonian rheologies. We show that rescaling the staggered grid stencil to better describe veloc-

ity components parallel to the fluid-solid interface significantly improves the accuracy. The code is verified using analytical solutions and then used to perform the permeability computations for a digital Fontainebleau sandstone sample (Andrä et al., 2013b).

## 2   Fluid flow in porous media

Fluid flow in porous media can be characterized with the Reynolds number which relates inertial to viscous forces:

$$Re = \frac{\rho v L}{\eta}, \tag{1}$$

where $\rho$ is fluid density, $v$ is velocity in direction of the flow, $L$ is the characteristic length and $\eta$ the fluid viscosity. Due to the small pore size, flows in porous media commonly exhibit small Reynolds numbers and are thus considered to be laminar

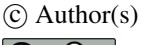


(Bear, 1988). For geological applications, Reynolds numbers typically are around $10^{-9} - 10^{-10}$ for magmas (Glazner, 2014) and range from $10^{-8}$ to $10^{-5}$ for ground water flow. This allows to simplify the incompressible Navier-Stokes equations to the Stokes equations (ignoring gravity):

$$\frac{\partial v_i}{\partial x_i} = 0 \tag{2}$$

$$\frac{\partial}{\partial x_j}\left[\eta\left(\frac{\partial v_i}{\partial x_j} + \frac{\partial v_j}{\partial x_i}\right)\right] - \frac{\partial P}{\partial x_i} = 0 \tag{3}$$

where $P$ denotes pressure, $v$ the velocity component and $x$ the spatial coordinate in Einstein summation convention.

If the pore structure of a porous medium is known, eq.(2) and eq.(3) can be used to directly model laminar fluid flow within this medium. However, at larger scales direct numerical simulation of porous flow is not feasible. In the case of Newtonian fluids, it is common to define a permeability $k$ which relates the flow rate $Q$ to the applied pressure gradient $\Delta P/L$ as well as fluid viscosity $\eta$:

$$k = -\frac{\eta L Q}{\Delta P A}, \tag{4}$$

where $A$ is the cross-sectional area of the porous medium. Eq.(4) is also known as Darcy's law and forms the basis of an effective description of Newtonian fluid flow in porous media (Andrä et al., 2013b; Saxena et al., 2017; Bosl et al., 1998). As stated above, this permeability is commonly determined by experimental methods on all scales. With the advent of numerical models for subsurface fluid flow (e.g. FEFLOW (Diersch, 2013)), it has become possible to predict large scale subsurface fluid flow using micro permeabilities as input parameter. Therefore an accurate prediction of micro permeabilies is necessary.

One possibility to do this is to relate the porosity $\phi$ of the medium to its permeability $k$. Deriving the exact nature of this relationship it not trivial and has been subject to a significant amount of research (Kozeny, 1927; Carman, 1937, 1956; Mavko and Nur, 1997). Due to the strong dependency of the permeability not only on porosity, but also on the 3D structure of the pore space, these approaches still suffer from inaccuracies. Due to the development of pore-scale numerical models, it has become possible to determine and refine the porosity-permeability relationship using direct numerical simulation on the basis of computed tomography (CT). These simulations typically provide solutions for fluid velocity $v$ and pressure $P$ for a given pressure gradient across the sample. From the velocity field in $z$- direction the volume-averaged velocity component $v_m$ is calculated (e.g. Osorno et al., 2015):

$$v_m = \frac{1}{V_f} \int\limits_{V_f} |v_z| \, dv, \tag{5}$$

where $V_f$ is the volume of the fluid phase. Making use of eq.(4) and $Q = v_m \cdot A$, the intrinsic permeability $k_s$ of the sample can then be computed as:

$$k_s = \frac{\eta v_m L}{\Delta P} \tag{6}$$





As described above, the flow of non-Newtonian fluids through porous media has gained considerable attention in recent years. Here, we use a power law rheology given by:

$$\eta = \begin{cases} \eta_1, & \text{if } \dot{\varepsilon} < \dot{\varepsilon}_1 \\ \eta_0 \left( \frac{\dot{\varepsilon}}{\dot{\varepsilon}_0} \right)^{n-1} \\ \eta_2, & \text{if } \dot{\varepsilon} > \dot{\varepsilon}_2 \end{cases} \tag{7}$$

where $\eta_1$ and $\eta_2$ are the upper and lower cutoff viscosities at the corresponding strain-rates $\dot{\varepsilon}_1$ and $\dot{\varepsilon}_2$. $\eta_0$ is the fluid viscosity at the reference strain-rate $\dot{\varepsilon}_0$ and $\dot{\varepsilon} = \sqrt{\frac{1}{2} \dot{\varepsilon}_{ij} \dot{\varepsilon}_{ij}}$ the effective strain-rate. $n$ is the power law exponent. With the definition adopted here, fluids with $n < 1$ are called shear-thinning, while fluids with $n = 1$ behave as Newtonian fluids and $n > 1$ are considered shear-thickening fluids. Note that this definition of $n$ differs from the common definition used in geodynamical modelling (called $n'$ here), where $n' = n^{-1}$.

In the case of non-Newtonian fluids, the definition of a permeability is not as straightforward as in the Newtonian case. Several studies have attempted to describe porous media permeability for non-Newtonian fluid rheologies. Until now a general description could not be found as used approaches differ. To develop a nonlinear variant of Darcy's law, Bird et al. (1960) assumed that porous media can be represented by parallel pipes and scaled up these capillary models to general porous media. By doing so, he suggested that the average velocity $v_m$ scales as a function of the driving force $F$ or the pressure gradient $\Delta P / L$ (Bird et al., 1960; Larson, 1981):

$$v_m = \left( \frac{k}{\eta_{eff}} \frac{\Delta P}{L} \right)^{\frac{1}{n}} = K_F \left( F \right)^{\frac{1}{n}} \tag{8}$$

where $k$ is the permeability, $\eta_{eff}$ an effective viscosity and $K_F$ a related model parameter. If $n = 1$ and $\eta_{eff} = \eta$, eq.(4) is recovered. Both the fraction $k/\eta_{eff}$ as well as $K_F$ depend on porosity $\phi$, stress exponent $n$, the reference viscosity $\eta_0$ and the pore scale geometry of the medium. Consequently, a simple expression for the permeability $k$ has not been found yet. Attempts to generalize Darcy's law based on eq.(8) include effective medium theories (Sahimi et al., 1990), pore network models (Shah and Yortsos, 1995) and pore-scale numerical simulations (Aharonov and Rothman, 1993; Vakilha and Manzari, 2008). Irrespective of the chosen approach and the exact form of either $k/\eta_{eff}$ or $K_F$, eq.(8) implies that a logarithmic plot of $v_m$ vs. either $\Delta P / L$ or $F$ should produce a straight line with slope $1/n$.

## 3 Method

We solve the system of governing equations (2) and (3) on a cubic lattice using the finite difference code LaMEM, which has originally been developed to simulate large scale deformation of the Earth's lithosphere and mantle (Kaus et al., 2016). Here, we will focus on modeling the flow of a fluid with both linear and non-linear viscosity $\eta$ through a rigid porous matrix. LaMEM employs a staggered grid finite difference scheme (Harlow and Welch, 1965) to discretize the governing equations (fig.1). Pressures are defined in the middle of the staggered grid cell, whereas velocities are defined on cell faces. Based on the





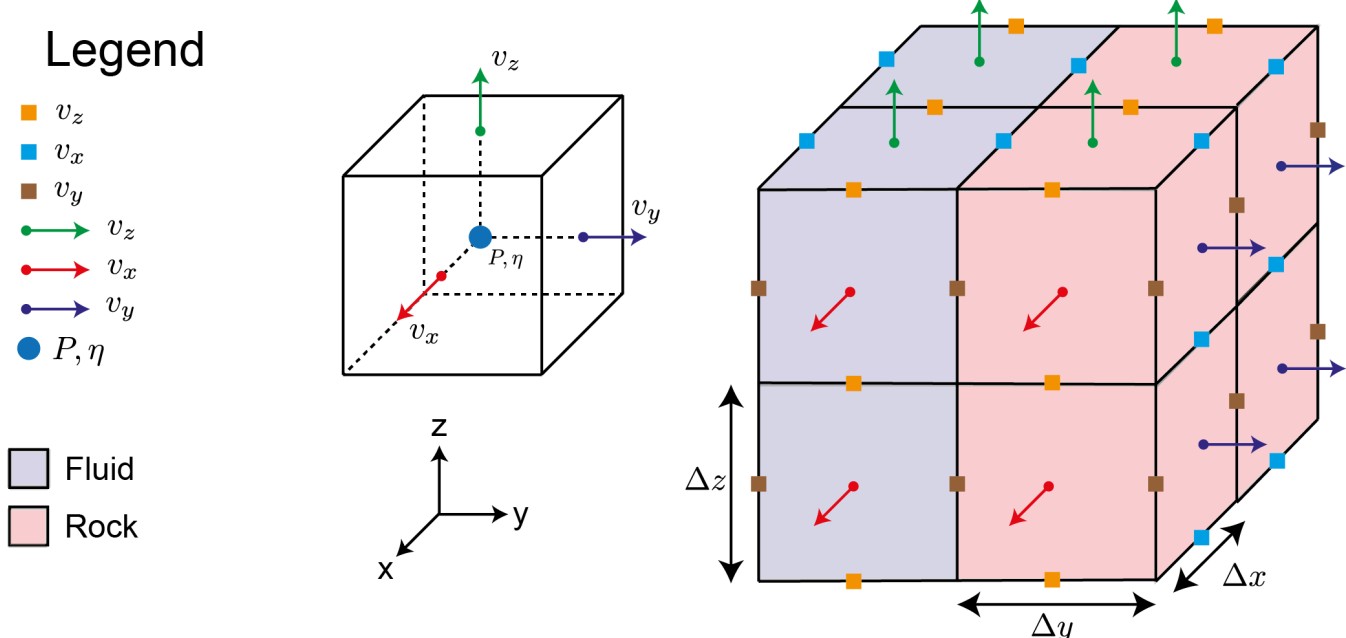

**Figure 1.** Staggered grid and location of variables.

data from CT-scans, each cell is assigned either a fluid or a solid phase. The discretized system is then solved using an iterative multigrid scheme to obtain values for velocities $v$ and pressure $P$. To this end, we employ multigrid solvers which are part of the PETSc library (Balay et al., 2010). As only cells belonging to the fluid phase exhibit non-zero values for the velocity, the velocity components belonging to solid cells are directly set to zero and only considered as boundary conditions. This greatly

reduces the degrees of freedom of the system to be solved and hence also the computational cost. To solve the linear system of equations a V-cycle geometric multiplicative multigrid solver is used (Fedorenko, 1964; Wesseling, 1995). The multigrid solver operates on up to five multigrid levels depending on the given input model. Convergence criteria are given by a relative convergence tolerance of $10^{-8}$ and an absolute convergence tolerance of $10^{-10}$ (see appendix A1). The absolute convergence tolerance $atol$ is defined as the absolute size of the residual norm and $rtol$ the decrease of the residual norm relative to the

norm of the right hand side. Therefore convergence at iteration $k$ is reached for:

$$\|r_k\|_2 < \max(rtol \cdot \|b\|_2, atol), \tag{9}$$

where $r_k = b - Cx_k$ with $b$ is the right-hand-side vector, $x$ the solution vector of the current timestep $k$ and $C$ the matrix representation of a linear operator (Balay et al., 2010).

Assigning solid and fluid phases to different cells defines the location of the fluid-solid interface. In the case of a staggered

grid, the location of the interface therefore does not correspond to the location of the interface-parallel velocity component. This is illustrated in fig.2 for the case of a $yz$-slice through the cube shown in fig.1. In this case, there is a vertical interface

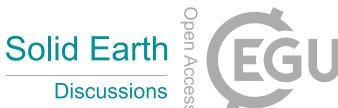



between two nodes. For constant grid spacing in $y$-direction the standard finite difference discretization of $\partial v_z / \partial y$ using eq.(3) reads as:

$$\frac{v_z^{(i+1,j,k)} - v_z^{(i,j,k)}}{\Delta y} \tag{10}$$

where the velocities at nodes $i$ and $i+0.5$ are computed and the velocity at $i+1$ is fixed to zero. Due to the misalignment of this velocity nodes with the fluid-solid interface, $v_z$ is not exactly zero. For this reason, the FD stencil is rescaled if it is located at the interface by using a non-central finite difference discretization as follows:

$$\frac{v_z^{(i+1,j,k)} - v_z^{(i,j,k)}}{0.5\Delta y} \tag{11}$$

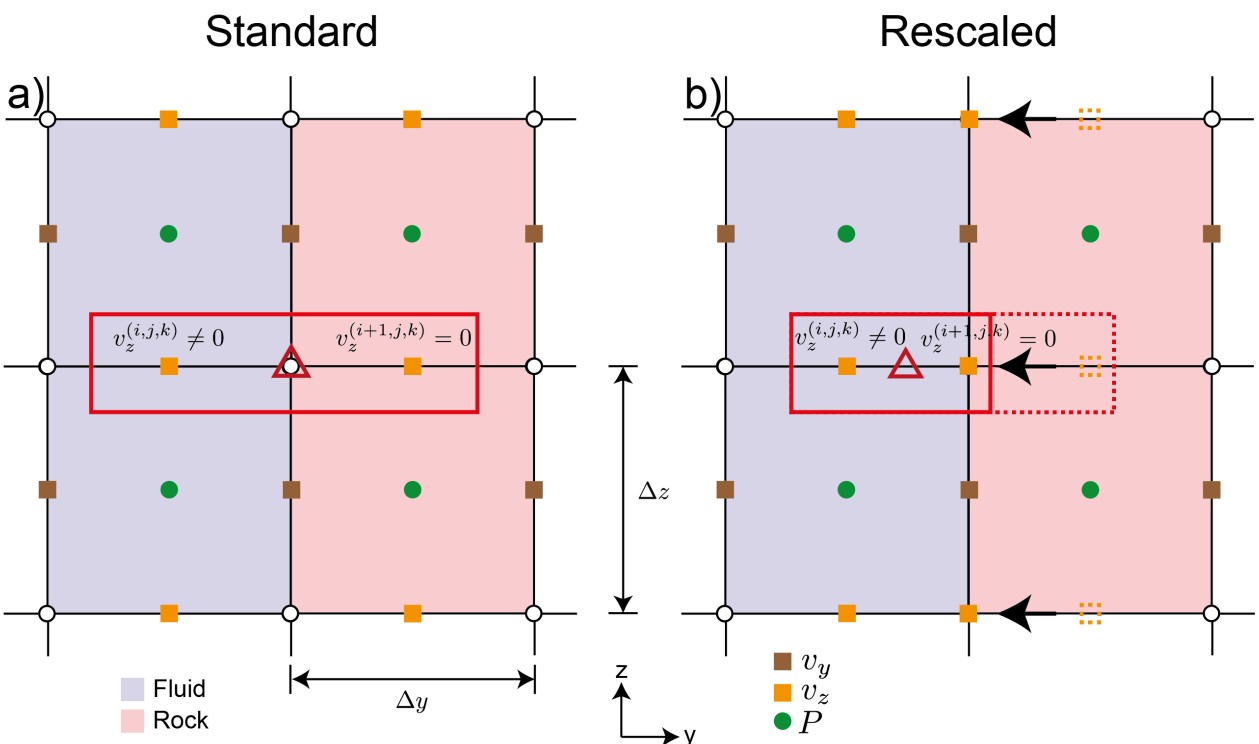

**Figure 2.** Staggered grid stencil rescaling. **a)** Stencil without rescaling and **b)** with stencil rescaling. The sketch is based on a cross section of Fig.1 in the $y-z$ plane.




# 4   Comparison with analytical solutions

To verify the method presented above, we performed a series of benchmark tests where we compared numerical solutions of simplified model setups to their respective analytical solutions. For simplicity, we non-dimensionalized the governing equations (2) and (3) as well as the rheology given in eq.(7) with characteristic values for viscosity $\eta_c$, length $l_c$, stress $\tau_c$ and velocity $v_c$:

$$\eta = \eta_c \cdot \tilde{\eta} \qquad\qquad x_i = l_c \cdot \tilde{x}_i \qquad\qquad (12)$$

$$\tau = \tau_c \cdot \tilde{\tau} \qquad\qquad v_i = v_c \cdot \tilde{v}_i = \frac{l_c \tau_c}{\eta_c} \cdot \tilde{v}_i, \qquad\qquad (13)$$

where the characteristic value for $v_c$ can be derived from the other characteristic values. Non-dimensional values are denoted with a ˜. For the remainder of this section, we will only use non-dimensional values and drop the ˜ for simplicity. Benchmark tests are organized as follows: first, we will present three benchmark test for the flow of a Newtonian fluid through i) a single tube, ii) multiple tubes and iii) through a simple cubic sphere pack, which is followed by a benchmark test of power law fluid flow through a single tube. The difference between numerically and analytically computed permeabilities is then expressed using the $L_2$ norm of their relative misfit:

$$\|\delta_k\|_2 = \sqrt{\left(\frac{k_{comp} - k_{ana}}{k_{ana}}\right)^2}, \qquad\qquad (14)$$

where $k_{comp}$ is the computed and $k_{ana}$ the analytically obtained permeability.

## 4.1   Newtonian flow through a single vertical tube

For a single vertical tube, the analytical solutions for both velocity $v$ and flow rate $Q$ are given as (e.g. Poiseuille, 1846; Landau and Lifshitz, 1987):

$$v = \frac{\Delta P}{4\eta L}(R^2 - r^2) \qquad\qquad (15)$$

$$Q = \frac{\pi \Delta P}{8\eta L}R^4, \qquad\qquad (16)$$

where $\frac{\Delta P}{L}$ is the pressure drop in $z$-direction, $R$ being the radius of the pipe and $r$ the integration variable. The characteristic scales in this case are given by $\eta_c = \eta_0$, $\tau_c = \Delta P$ and $l_c = R$ so that the pipe radius $R$, fluid viscosity $\eta$ and pressure difference $\Delta P$ all take values of 1. The cubic model domain has an edge length of 4 units. Combining eq.(16) with eq.(4), the non-dimensional permeability is then given by:

$$k = \frac{L \frac{\pi \Delta P}{8\eta L} R^4}{\Delta P A} = \frac{\pi}{128} \qquad\qquad (17)$$



To assess the effect of different spatial resolutions, we conduct a resolution test where we increase resolution from $8^3$ to $256^3$ nodes with a constant grid spacing in each direction. Four sets of resolution tests were conducted. In the first two sets, permeability was computed using the standard finite difference approach without stencil rescaling. The two sets then differ due to the exact location of the pipe. In set 1, the location of the pipe was chosen in such a way that the pipe surface aligned with

the numerical grid (standard, ON NODE) so that computational nodes were directly located on the fluid/solid interface in in $x$- and $y$-direction. In set 2 (standard, OFF NODE), the location of the pipe was shifted so that the fluid/solid interface was located between the respective computational nodes. The same procedure was applied to sets 3 and 4 where stencil rescaling was employed. The reason to do that was to determine the effect of well-aligned computational nodes, as this is often not the case in more complex geometries.

As expected, the numerical results generally show higher accuracy when stencil rescaling is employed and when node locations and interfaces of the tube are aligned (see Fig.3). The order of convergence is linear for cases without rescaling or when the tube interface does not coincide with grid nodes, but superlinear if both rescaling is employed and interface and node location coincide.

### 4.2   Newtonian flow through multiple vertical tubes

In natural rocks larger channels tend to dominate the overall permeability. To assess this effect, we compute the flow through several straight tubes with different radii (Fig.4). We use four pipes with non-dimensional radii given as $R_1 = 1$, $R_2 = 2$, $R_3 = 4$, $R_4 = 8$. The viscosity of the fluid is 1 and edge length of the cubic domain is 8. The simulations are performed in a similar manner as the single tube benchmark by increasing the number of grid points from $8^3$ to $256^3$. For each tube the analytical solution (eq.(15),(16)) is computed and the cumulative analytical permeability value is compared against computed

values. The non-dimensional permeability in this case reads as:

$$k = \frac{LQ(R_1^4 + R_2^4 + R_3^4 + R_4^4)}{\Delta PA} \tag{18}$$

The individual tubes contribute to the absolute permeability as follows: P1 = 0.3662 %, P2 = 0.0229 %, P3 = 93.7514 %, P4 = 5.8595 %.

Similar as observed for the single tube setup, the results show a lower relative error for calculations employing the stencil

rescaling compared to those without. Furthermore as shown for the setups with single tube the results are more accurate in cases where the numerical grid aligns with the tube surface. As expected, the overall permeability is dominated by the largest tube, as we do not see any significant changes within the relative error of the computed permeability.

### 4.3   Newtonian flow through simple cubic (SC) sphere packs

In order to verify the code for more complex geometries as the vertical tube, we here consider simple cubic (SC) sphere packs.

Sphere packs provide a geometry for different packings as the porous medium is homogeneous. The setup has dimensions of 2 in all directions.



**Figure 3.** Hagen-Poiseuille benchmark results. Shown is the error norm $\|\delta_k\|_2$ vs. spatial resolution. The different curves show cases where the tube surface coincides with a nodal point (ON NODE) or not (OFF NODE). Blue lines represent simulations using stencil rescaling, whereas red lines denote simulations without stencil rescaling.

The permeability of an SC sphere pack is given by (Bear, 1988):

$$k = \frac{\phi^3 \cdot d_{sp}^2}{180 \cdot (1 - \phi)^2} \tag{19}$$





**Figure 4.** Multiple tube Hagen-Poiseuille benchmark. Lines and symbols correspond to the same cases as in fig.3.

where $d_{sp}$ is the sphere diameter and $\phi$ is the porosity for simple cubic packing of $1 - \frac{\pi}{6} \approx 0.476$, respectively.

Fig.5 shows the increase in accuracy with increasing number of grid points employed. The presented relative errors of the permeability value is computed in the same manner as shown in eq.(14). The simulations employing stencil rescaling show a better convergence and seem to saturate against an relative error of $10^{-1}$, demonstrating the influence of boundary effects through applied no-slip boundary conditions (finite size effect).



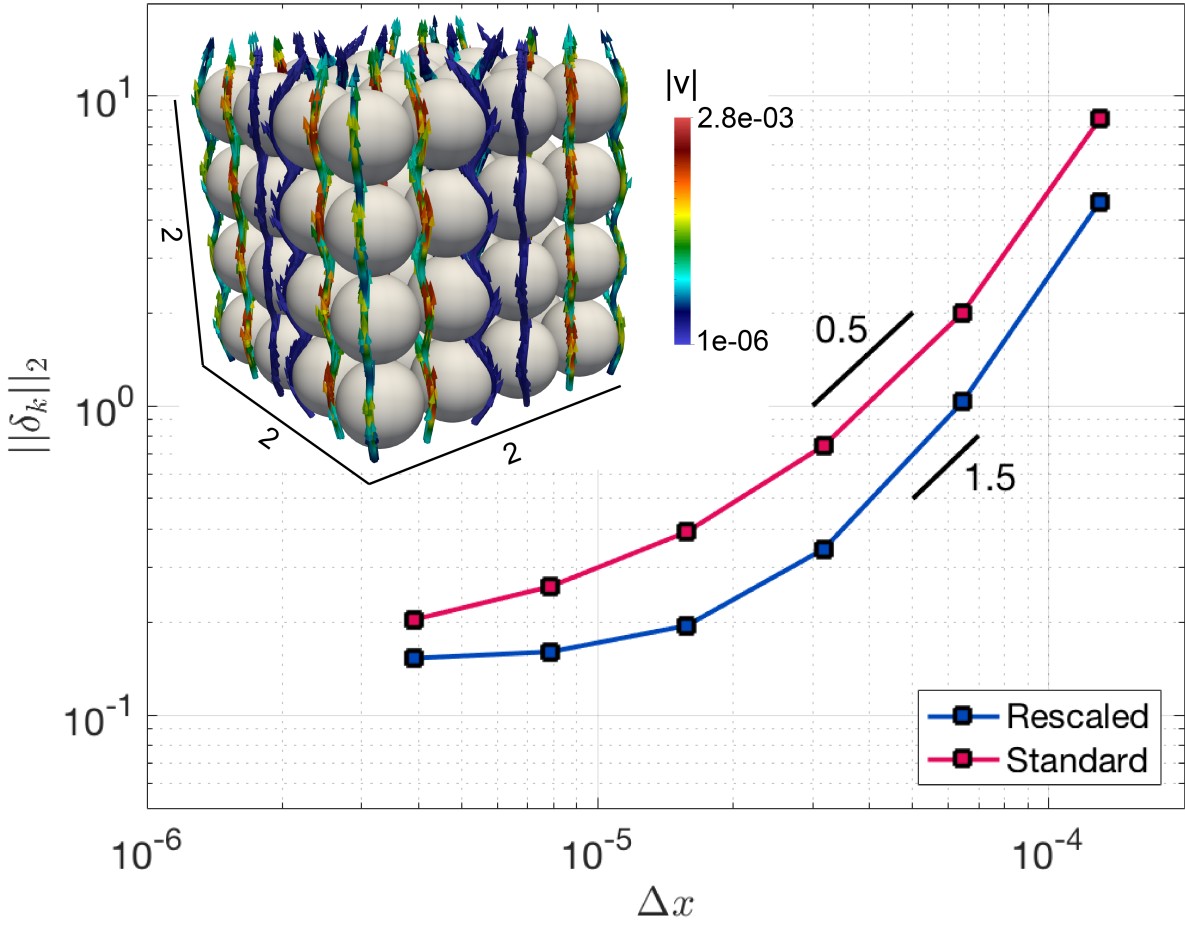

**Figure 5.** Computed $||\delta_k||_2$ norm of the misfit between analytically and numerically computed permeabilities. The inset shows the discretization using 4 spheres in each direction (64 spheres in total). Streamlines are computed around those spheres and colorized with the computed velocity. Blue dots show results using stencil rescaling and red dots results with the standard method.

### 4.4 Power law fluid flow through a single vertical tube

In order to verify the computed value we compare this setup against an analytical solution of Hagen-Poiseuille flow with power law fluid behaviour. For the single tube configuration described in sec.4.1 and a power law rheology, the velocity within the tube is given by (e.g. Turcotte and Schubert, 2002):

$$5 \quad v_z(r) = \frac{C_1}{\frac{1}{n}+1} \cdot \left(\frac{\Delta P}{L}\right)^{\frac{1}{n}} \cdot \left(\left[\frac{R}{2}\right]^{\frac{1}{n}+1} - r^{\frac{1}{n}+1}\right), \tag{20}$$



where $C_1 = 2\eta_0^{-\frac{1}{n}}$ (see Appendix B), $R$ is the tube radius and $r$ the width of the tube in Cartesian coordinates. Fig.6 shows a good agreement between the numerical and analytical velocities for non-Newtonian fluids using power law exponents ranging from $0.5$ to $2$, covering most fluids used for enhanced oil recovery (e.g. Najafi et al., 2017; Xie et al., 2018).

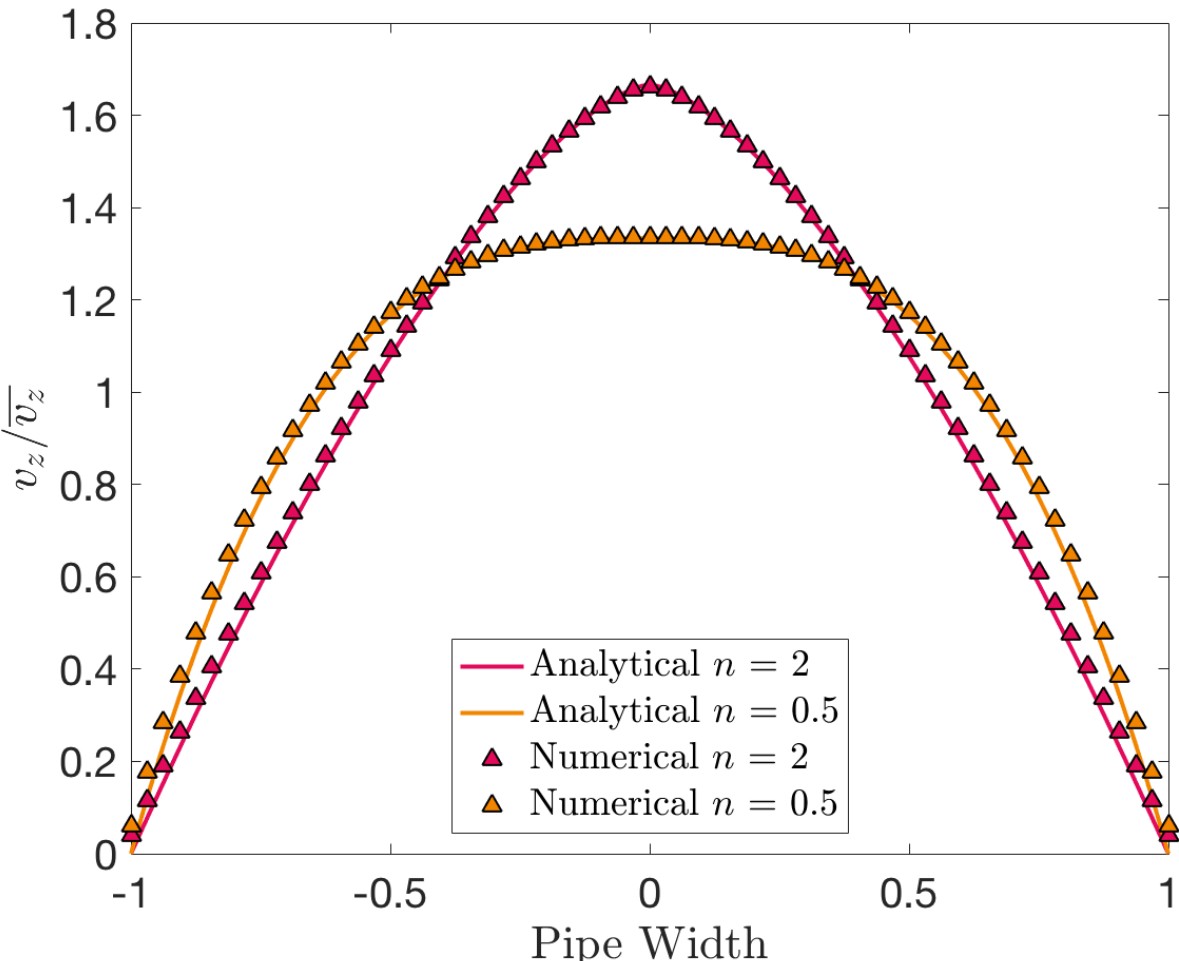

**Figure 6.** Comparison of analytical and numerical velocities for Hagen-Poiseuille flow with a power law fluid. Analytical velocities are represented as colored lines and numerical velocities as colored symbols.

## 5 Application to Fontainebleau sandstone

5    To verify the ability of the code to handle more complex flows through natural samples and to validate previously computed permeability values we used the CT data for a Fontainebleau sandstone sample provided by Andrä et al. (2013b) with dimen-





sions 2.16 $mm$ × 2.16 $mm$ × 2.25 $mm$ (resolved with 288×288×300 grid points). In order to optimize the computation and reduce computational resources a subsample with dimensions of $256^3$ is used for further computations. The sample mainly consists of monodisperse quartz sand grains and is therefore a very popular sample for numerical and experimental permeability measurements. Furthermore sandstone is known to be a ideal reservoir rock and is of certain interest for several geological

fields, especially in exploration geology. Laboratory measurements of the given sample with porosity ≈ 15.2 % result in a permeability value of ≈ 1100 $mD$ (Keehm, 2003).

## 5.1  Newtonian flow

According to previous tests we compute a Newtonian permeability of the extracted subsample by using eq.5 and 4. Fig.7A shows streamlines colored using computed fluid velocities and Fig.7B the applied pressure. Knackstedt et al. (2009); Zhang

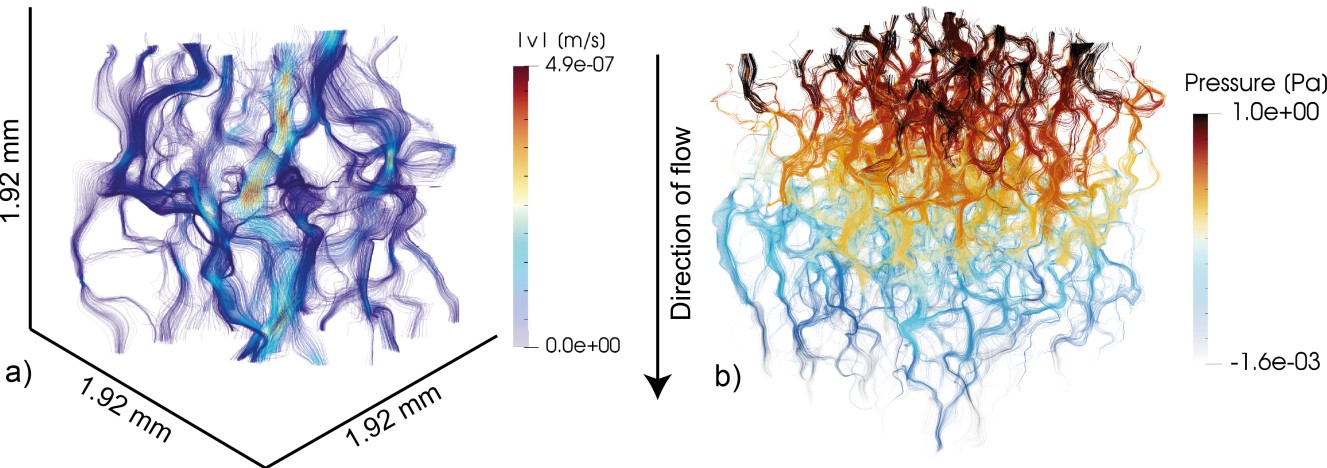

**Figure 7.** Newtonian fluid flow through the Fontainebleau sandstone sample. Streamlines colored using computed fluid velocities are shown in **a)** and streamlines colored using fluid pressures are shown in **b)**.

et al. (2000) showed that e.g. Lattice-Boltzmann method overestimates permeability values using coarse grid resolutions. This effect might also be present in Andrä et al. (2013b) as the computed permeability values, compared to laboratory values, tend to be overestimated as well. Yet this previous work did not demonstrate numerical convergence of the present solution. To test this effect further, we stepwise increased the resolution of the Fontainebleau subsample by a factor of 2, 3 and 4 ($512^3$,$768^3$,$1024^3$). Fig.8 shows a comparison between the computed and measured values for the given Fontainebleau dataset. With increasing

resolution of the subsample the computed permeability value seems to converge against the laboratory value. In comparison to the initial resolution of $256^3$ the computed permeability values decreased by ≈ 24.6 % when using a grid resolution of $1024^3$. Additionally the benefit of stencil rescaling can also be seen here, as e.g. the simulation with a resolution of $512^3$ and stencil rescaling predicts nearly the same permeability as the case with doubled resolution and no stencil rescaling. Clearly, the models





converge to a value that is close to the measured value. The numerical convergence is computed for several subsamples (see appendix C). Fig. 8 represents the convergence of a single subsample.

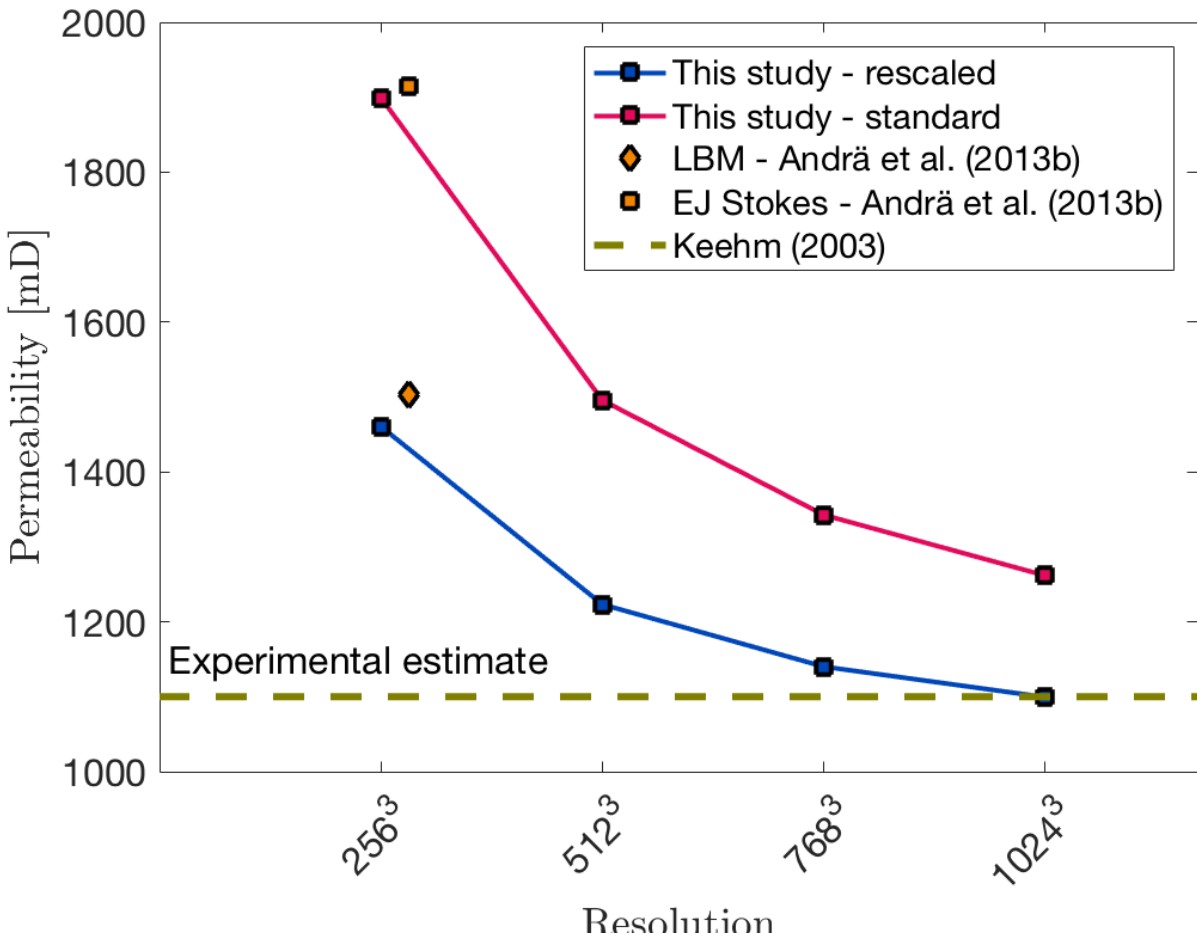

**Figure 8.** Computed permeability values against grid resolution. Orange symbols denote simulations using Lattice-Boltzmann method (LBM) and explicit jump stokes (EJ Stokes), both methods are used in Andrä et al. (2013b). Blue data points represent simulations using stencil rescaling while simulations represented by red dots use the standard method. Brown dotted line symbolizes the experimental estimate from Keehm (2003).

## 5.2 Power law fluid flow

To demonstrate the capability of the code to compute the flow of non-Newtonian fluids through porous media, we computed the average flow velocity $v_m$ for a square subsample of the Fontainebleau sandstone sample described above using the powerlaw rheology given in eq.(7). The edge length of the subsample was 1.92 mm, which corresponds to a CT resolution of $256^3$ voxels.



To increase accuracy, we increased this resolution by a factor of 2 to a resolution $512^3$. As seen in the section 5.1, results at this resolution may overestimate the actual permeability value. The chosen resolution thus represents a compromise between accuracy and computational cost. The reference viscosity was set to $\eta_0 = 1$ Pas and $\eta_1$ and $\eta_2$ were set to $10^{-3}$ and $10^6$ respectively. Two sets of simulations using a power law exponent of 0.5 and 1 were performed. In each set the applied pressure

5    at the top boundary is changed from 1 - 16 Pa. In fig.9 we plot the applied pressure at the top boundary against the computed average velocity. For both sets of simulations the computed slopes of $1.998 \pm 8.668 \times 10^{-4}$ and $1.000 \pm 2.582 \times 10^{-6}$ are in good agreement with the imposed power law coefficients of 0.5 and 1 (eq.(8)).

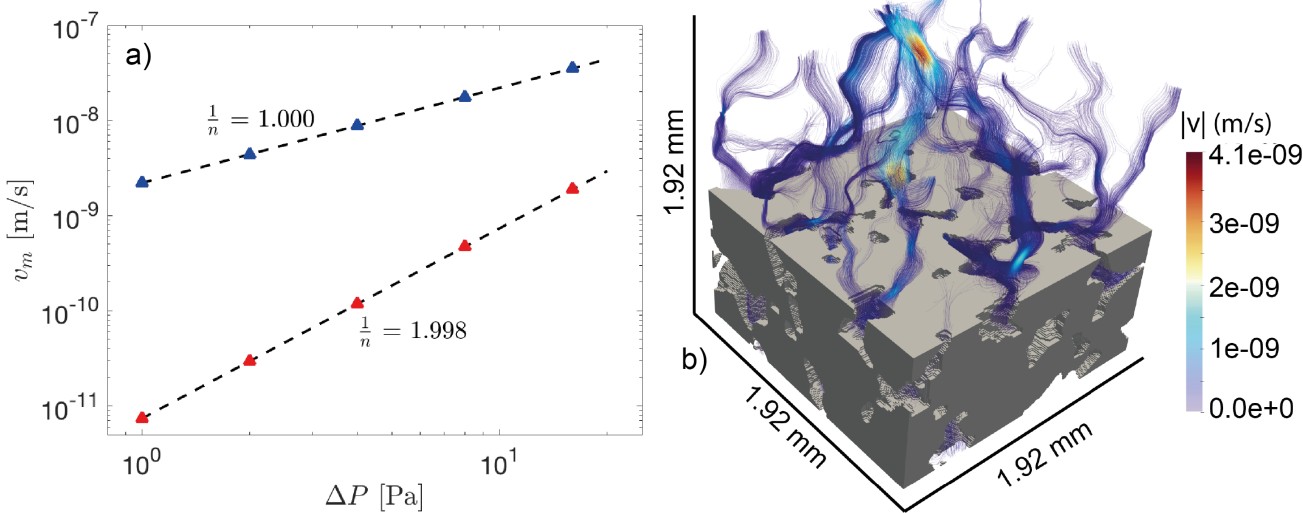

**Figure 9.** Computed results on the Fontainebleau sample using non-Newtonian rheology. **a)** shows the mean velocity against the applied pressure at the top boundary. Red and blue triangles symbolize each simulation and the corresponding dotted black line represents the fitted curve through the obtained data with slope $\frac{1}{n}$. **b)** illustrates computed streamlines of the Fontainebleau subsample using a power law coefficient of 0.5. Solid material is displayed in grey and the streamlines are colored according to computed velocities.

## 6    Discussion and Conclusion

In this paper, we presented the capability of the open-source finite difference solver LaMEM to compute the permeability of
10    given porous media. The code was verified using a set of benchmark problems with given analytial solutions ranging from Hagen-Poiseuille flow through vertical tubes to more complex flow through simple cubic sphere packs. Using CT Data of a Fontainebleau sandstone, we then demonstrated that the code is able to predict the permeability of natural porous media. In both benchmarks and application tests, the benefits of the stencil rescaling method can be observed, as this method provides significantly more accurate results at no additional computational cost.



Benchmarking results for single and multiple tubes demonstrate that the permeability calculation improves considerably in case the fluid-solid interface and the numerical grid are at least partially aligned. Cases using the stencil rescaling solutions with a velocity change on a computational node produce smaller relative errors.

Similar to studies using Lattice-Boltzmann method (Knackstedt and Zhang, 1994; Zhang et al., 2000; Keehm, 2003) our res-
olution test for the Fontainebleau subsample shows that the computed permeability value also decreases with increasing grid resolution. For instance, computing the permeability of Fontainebleau sandstone sample with grid resolution of $1024^3$, calculations employing stencil rescaling give approximately the same permeability value as suggested by laboratory measurements, while simulations without employing stencil rescaling overestimate the computed permeability by $\approx 14.72\,\%$. (Fig.8).

The computation of permeabilities in a three dimensional pore space using micro-CT data strongly depends on a reasonable
quality of the micro-CT images followed by several steps of segmentation in order to resolve tiny fluid pathways. Although high quality input data is required in most cases it is usually computationally expensive to use the entire micro-CT scan with full resolution, thus representative subvolumes or a reduced numerical resolution has to be used as computational resources are limited.

Additionally the segmentation of the CT data has a considerable effect on the computed permeability as discussed in Andrä
et al. (2013a), since segmentation of the acquired micro-CT data has a major effect on the three dimensional pore space and therefore on the obtained value. In two phase systems (fluid/solid), segmentation is straightforward whereas it may become more difficult in multiphase systems. All of the above points are a source of uncertainty and need to be considered when comparing numerical calculations to laboratory measurements for rock samples. Furthermore we showed that LaMEM is able to compute non-Newtonian fluid flow in porous media, which is not only relevant for geosciences but also of importance for
industrial applications (Saidur et al., 2011).

Furthermore it should be kept in mind that solver options like convergence criteria may influence the obtained permeability result. Fig.A1 (see Appendix A) highlights the effect of different relative tolerances on the computed permeability value. In order to avoid spurious results, we recommend to test the influence of the relative and absolute tolerance on the model outcome.

In conclusion the capability of the open-source finite difference solver LaMEM to accurately simulate Newtonian and non-
Newtonian fluid flow in porous media is successfully demonstrated for different setups with an increasing geometric complexity including pipe flow, ordered sphere packs and a micro-CT dataset of Fontainebleau sandstone.

## Appendix A: Convergence criteria

To determine whether a numerical solution converges, two convergence criteria are used, which are absolute and relative convergence tolerance. To test the effect on the numerical solution we varied both while computing permeability of three
different setups. Our results show that the obtained permeability value saturates for relative convergence tolerances $< 10^{-7}$. Thus for all further simulations a relative convergence tolerance of $10^{-8}$ is used (Fig.A1). A change in the absolute convergence tolerance did not have any effect on the computed solution, therefore we use a absolute convergence tolerance of $10^{-10}$.





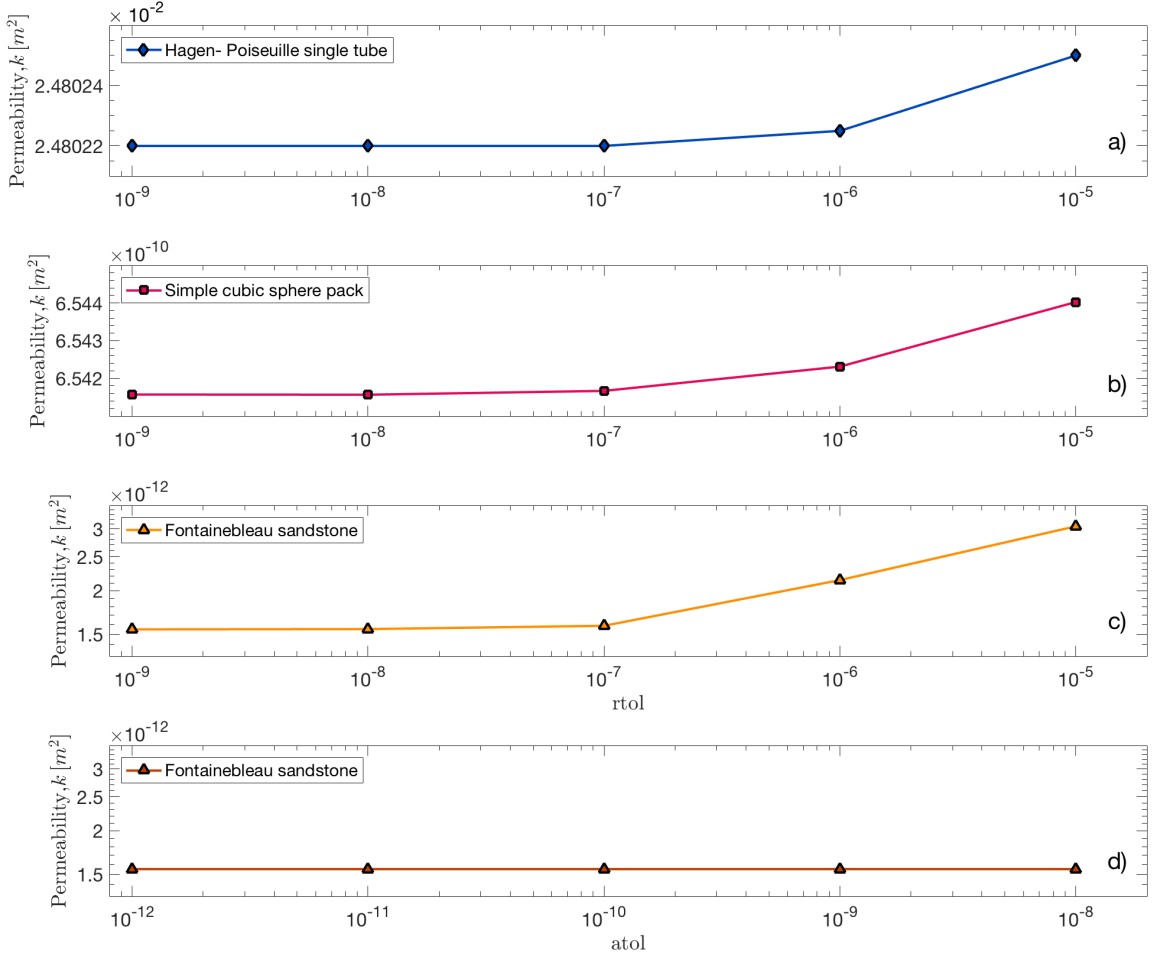

**Figure A1.** Results of simulations for (a) Hagen-Poiseuille single tube, (b) simple cubic sphere pack and (c)+(d) Fontainebleau sandstone using different relative/absolute convergence tolerances.

## Appendix B:  Definition of $C_1$

$C_1$ is an constant arising during the derivation of eq.(20), which is related to the nonlinear rheology used in Turcotte and Schubert (2002). This rheology is written as:

$$\dot{\epsilon} = C_1 \tau^{n'} \tag{B1}$$





where $n'$ is the stress exponent as used for powerlaw materials in geodynamics. Replacing $\tau$ with $\tau = 2\eta\dot{\epsilon}$ leads to:

$$\dot{\epsilon} = C_1(2\eta\dot{\epsilon})^{n'} \tag{B2}$$

Solving eq.B2 for $\eta$ results in:

$$\eta = \frac{1}{2}C_1^{-\frac{1}{n'}}\dot{\epsilon}^{\frac{1}{n'}-1} \tag{B3}$$

We can now define a reference viscosity $\eta_0$ at a reference strain rate $\dot{\varepsilon}_0$. This reference viscosity then reads as:

$$\eta_0 = \frac{1}{2}C_1^{-\frac{1}{n'}}\dot{\epsilon}_0^{\frac{1}{n'}-1} \tag{B4}$$

Assuming $\dot{\epsilon}_0 = 1$ and solving for $C_1$ then provides us with the following expression:

$$C_1 = 2\eta_0^{-n'} = 2\eta_0^{-\frac{1}{n}} \tag{B5}$$

## Appendix C: Permeabilities of different Fontainebleau subsamples

In order to show numerical convergence of the given Fontainebleau sample several subsamples were extracted and the resolution increased to $512^3$, $768^3$ and $1024^3$ grid points. Fig. C1 displays the convergence with increasing grid resolution. The different subsamples show a variance of around $12\%$ for the computed permeability value.

*Author contributions.* PE did the bulk of the work, including writing, visualization, methodology and running simulations. MT and GJG designed the study and contributed to manuscript writing. AP implemented stencil rescaling into LaMEM. WF assisted in code benchmarking.

OK performed the resolution test for Fontainebleau subsamples. BJPK contributed to data interpretation and manuscript writing.

*Competing interests.* The authors declare that they have no competing interests.

*Acknowledgements.* This work has been founded by DFG project International Research Training Group 2156 (IRTG) Deep Earth Volatile Cycles and by BMBF GEON project PERMEA. M.T. has received funding from the Bayerisches Geoinstitut Visitors Program. Simulations were performed on the btrzx2 cluster, University of Bayreuth and the Mogon II cluster, Johannes Gutenberg University, Mainz.

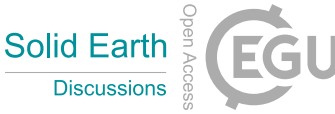



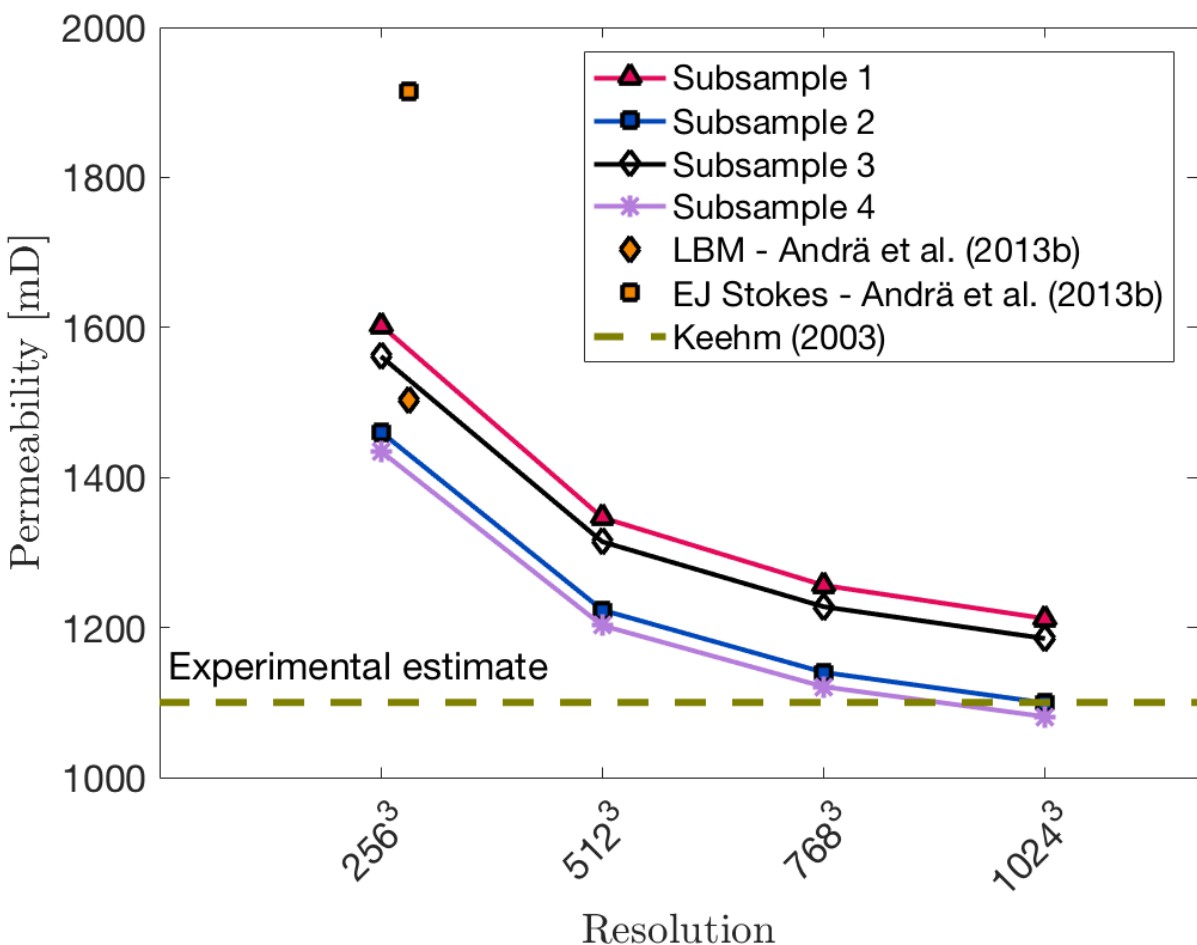

**Figure C1.** Numerical convergence of different Fontainebleau subsamples with increasing grid resolution. All subsamples displayed were computed using stencil rescaling. For comparison the resulting computed permeability from Andrä et al. (2013b) are shown. The dotted brown line symbolizes the experimental estimate taken from Keehm (2003).



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
