# Peer review of "Pore-scale permeability prediction for Newtonian and non-Newtonian fluids"

_Solid Earth, 2019_

## Referee Comment (RC1) · Kirill Gerke (Referee) · 30 Jun 2019

The paper describes a modification of the existing codebase to simulate flow of New-tonian and non-Newtonian fluids on the pore level. I found the paper to be sound and scientifically robust. It is very well-structured and writing is also good.

For aforementioned reasons i strongly recommend the paper for publication: it presents new code (which is probably more accurate that other existing FDM alternatives), it presents a rescaling technique that help to improve the quality of the solution, it presents a realization for non-Newtoniam fluids. The manuscript is good 'as is', but i still recommend the Authors to consider following changes before submitting their very final version of the paper:

[Figure]

1) I guess as the code is the part of the LaMEM now, it should be open source, isn't it? If so, please, provide a link to the repository somewhere at the relevant part of the paper.

2) Within your abstract and introduction you mention that non-Newtonian code is necessary for nano-fluids and some related problems. I would suggest a couple of sentences to explain this a bit, because technically you provide a solution for micro-scale. I would also guess that magma flow is a potential object of simulations with your code.

3) Equation 5 is a technically valid for any flow direction, not sure why do you talk about z-direction here. I would suggest re-writing it for the general case, especially considering that later on in Eq.6 you do you generalized form to compute permeability.

4) Something went wrong with Eq.12-13 (probably while converting to pdf?). Please, fix these.

5) Could not completely catch the meaning of all elements on Fig.3. You have black lines with attached numbers of 3 and 0.25 (the latter is partially covered by the tube flow figure inset). Please, consider fixing this.

6) I found a disagreement between Eq.18 and Fig.4 - in the text you assign R3=4, R4=8, while the pipe #3 is larger on the figure, plus you report that #3 contributed more to the flow. Seems like you interchanged #3 and #4 at some point.

7) Fig.5 - you have quite slow (blue) flow lines at the same positions as higher (yellow-red) flow lines at the same locations along the flow direction. I find this to be somewhat strange, considering that the flow should be symmetrical around the spheres under periodic boundary conditions (you should use them, otherwise you can't compare against analytical solutions for drag forces).

8) Not 100% sure here, but i do not think that Eq.19 was derived by Bear, as analytical solutions for spheres (not only SC, but BCC and FC packings) comes from preceding papers, e.g.: Sangani, A.S., Acrivos, A., 1982. Slow flow through a periodic array of

spheres. Int. J. Multiph. Flow 8, 343–360. doi:10.1016/0301-9322(82)90047-7

9) Fig.8 and the text related to this figure. First, how did you produce those different resolution figures? From the results i would guess you simply "magnified" each voxel 2 times to consist of 4 voxel for each magnification step. Please, describe your methodology. Because i would expect somewhat different behavoir if you would scale your samples while conserving its spatial statistics: Karsanina, M. V., & Gerke, K. M. (2018). Hierarchical Optimization: Fast and Robust Multiscale Stochastic Reconstructions with Rescaled Correlation Functions. Physical Review Letters, 121(26), 265501. Now, you mention that LBM also converges from above and cite some papers with such behaviour. I guess these papers used single-relaxation LBM. Technically, LBM can converge from below, above, and from below and above at the same time. To improve this section of the text i recommend reading and citing the following papers: Khirevich, S., Ginzburg, I., & Tallarek, U. (2015). Coarse-and fine-grid numerical behavior of MRT/TRT lattice-Boltzmann schemes in regular and random sphere packings. Journal of Computational Physics, 281, 708-742. Khirevich, S., & Patzek, T. W. (2018). Behavior of numerical error in pore-scale lattice Boltzmann simulations with simple bounce-back rule: Analysis and highly accurate extrapolation. Physics of Fluids, 30(9), 093604. Zakirov, T., & Galeev, A. (2019). Absolute permeability calculations in micro-computed tomography models of sandstones by Navier-Stokes and lattice Boltzmann equations. International Journal of Heat and Mass Transfer, 129, 415-426.

10) I would recommend to present a very brief comparison against existing FDM codes, for example FDMSS. I would expect that your code is more accurate, yet takes much longer time to converge and more computationally heavy in terms of CPU and RAM.

All in all, i think this work to be of very good quality and ready to be published with Solid Earth. My comments are more "cosmetic" in this regard.
* * *

---

## Referee Comment (RC2) · Stephane Beaussier (Referee) · 17 Jul 2019

The manuscript proposes a new application of the LaMEM open-source code - originally designed for geodynamic applications - to the modelling of sample-scale newtonian and non-newtonian fluids in porous systems. The main application proposed by the author is the estimation of rock permeability as a substitute/complement to laboratory measurement.
This study is meaningful tool to tackle the long running problem of estimating permeability of rocks. In particular, when laboratory measurements are difficult or impossible. It brings significant improvement compared to other numerical models by introducing stencil rescaling along rock-fluid boundaries which seem to greatly improve the accuracy of the permeability estimation while limiting the computational cost. The

robustness of the code is backed by sufficient benchmarking as well as comparison with experimentally measured permeabilities. Furthermore, the manuscript is clear and well written. Overall I would recommend this manuscript for publication in its current form with only a few minor – mostly cosmetic - modifications. Here is a list of minor suggestions to improve the manuscript:

1. I could not find any link to this manuscript version of the LaMEM code in the text. If I am correct it is an open source code and therefore it would be necessary to provide the code as an online supplementary or at least a link to the repository in the manuscript.

2. The only issue I have with this manuscript is that I find the information provided on the stencil rescaling a little limited. Given the importance it takes in the manuscript, I would expect a more extended explanation of the method and its implementation in the code. In particular, I believe better discussion on the stability an accuracy of the FD stencil rescaling with references would improve the quality of the manuscript.

3. In page 12 line 2 is written: "using power law exponents ranging from 0.5 to 2." Yet, in figure 6 you only show two values that are tested rather than a range. I would suggest changing the phrasing to "when using 0.5 and 2 as values for the power law exponents".

4. In Fig. 3, 4 and 5 black lines with a numerical value are shown but not explained in the caption. It took me quite some time to understand it was the curve local slop. Therefore, I suggest adding a sentence in the caption to explicitly tell what these black lines are, or remove them as the figures are already self-explanatory without giving a numerical value for the local slop. item In figure 4 you most likely flipped P3 and P4 in the top left corner schematic as according to line 22-23 page 8 P4 is the largest tube and not P3.

5. In figure 5 the box displaying streamlines around the sphere show significant variations of flow velocity perpendicular to the direction of flow (3 orders of magnitude!!). This is very puzzling as I would expect the flow to be relatively homogenous laterally. Is this a rendering mistake or a consequence of boundary conditions? It should either be corrected or explained in the text.

6. In figure A1: rtol should be added below the x-axis of subfigure a) and b).

7. In the authors contribution there is a spelling mistake in the authors initials. OK is written instead of MOK for M. O. Kottwitz.

[Figure]

---

## Author Comment (AC1) · 21 Aug 2019

We thank Kirill Gerke for his review. His useful comments helped us to improve our manuscript.

Please find below a point by point response to the comments (comments of the reviewer in black and our response in blue) and the revised version in the supplement.

Sincerely, on behalf of the authors Philipp Eichheimer

1. I guess as the code is the part of the LaMEM now, it should be open source, isn't it? If so, please, provide a link to the repository somewhere at the relevant part of the paper.

   A link to the open-source repository as well as the revision number, which has been used to reproduce the results of this work, has been added. (Page 18, line 4)

2. Within your abstract and introduction you mention that non-Newtonian code is necessary for nano-fluids and some related problems. I would suggest a couple of sentences to explain this a bit, because technically you provide a solution for micro-scale. I would also guess that magma flow is a potential object of simulations with your code.

   We rewrote the section in the introduction to provide more information on nanofluids and magma flow. (Page 2, line 20-27)

3. Equation 5 is a technically valid for any flow direction, not sure why do you talk about z-direction here. I would suggest re-writing it for the general case, especially considering that later on in Eq.6 you do you generalized form to compute permeability.

   This is correct, but the version of LaMEM, used in this study, only computes a volume average z-velocity. This velocity is then used to compute permeability in z-direction using eq.(6). We therefore decided to leave eq.(5) and (6) as is.

4. Something went wrong with Eq.12-13 (probably while converting to pdf?). Please, fix these.

   We changed the equations to make sure they are displayed properly. (Page 8, line 1-4)

5. Could not completely catch the meaning of all elements on Fig.3. You have black

lines with attached numbers of 3 and 0.25 (the latter is partially covered by the tube flow figure inset). Please, consider fixing this.

The black lines show local slopes of the curve. In order to clarify this issue we added this information into the figure captions and moved the inset of figure 3.

6. I found a disagreement between Eq.18 and Fig.4 - in the text you assign R3=4, R4=8, while the pipe #3 is larger on the figure, plus you report that #3 contributed more to the flow. Seems like you interchanged #3 and #4 at some point.

Sorry for this mistake, we changed the the values on page 9 line 17 to fit the inset in figure 4.

7. Fig.5 - you have quite slow (blue) flow lines at the same positions as higher (yellow- red) flow lines at the same locations along the flow direction. I find this to be somewhat strange, considering that the flow should be symmetrical around the spheres under periodic boundary conditions (you should use them, otherwise you can't compare against analytical solutions for drag forces).

We changed the figure as the rendered streamlines were not representative and thus the figure was perhaps confusing. Figure 5 now shows computed streamlines of the velocity around the spheres. This should make it easy to understand and highlight the flow structure.

Concerning the boundary conditions we use free-slip at the side boundaries of the domain and no-slip at the internal solid-fluid interface. In the case of simple cubic systems the velocities at the boundary are symmetric and therefore the effect of boundaries on the result should be negligible. We added an additional sentence in Methods section to clarify the employed boundary conditions. (Page 5, line 15-16)

8. Not 100 % sure here, but i do not think that Eq.19 was derived by Bear, as analytical solutions for spheres (not only SC, but BCC and FC packings) comes
from preceding papers, e.g.: Sangani, A.S., Acrivos, A., 1982. Slow flow through a periodic array of spheres. Int. J. Multiph. Flow 8, 343–360. doi:10.1016/0301-9322(82)90047-7.

We added the reference of Sangani and Acrivos (1982) as they describe the flow through a periodic array of spheres for simple cubic packing. However, the exact expression used in our manuscript is not explicitly stated in Sangani & Acrivos, but rather in Bear (1988), which is why we kept both references.

9. Fig.8 and the text related to this figure. First, how did you produce those different resolution figures? From the results i would guess you simply "magnified" each voxel 2 times to consist of 4 voxel for each magnification step. Please, describe your methodology. Because i would expect somewhat different behavoir if you would scale your samples while conserving its spatial statistics: Karsanina, M. V., & Gerke, K. M. (2018). Hierarchical Optimization: Fast and Robust Multi-scale Stochastic Reconstructions with Rescaled Correlation Functions. Physical Review Letters, 121(26), 265501. Now, you mention that LBM also converges from above and cite some papers with such behaviour. I guess these papers used single-relaxation LBM. Technically, LBM can converge from below, above, and from below and above at the same time. To improve this section of the text i recommend reading and citing the following papers: Khirevich, S., Ginzburg, I., & Tallarek, U. (2015). Coarse-and fine-grid numerical behavior of MRT/TRT lattice-Boltzmann schemes in regular and random sphere packings. Journal of Computational Physics, 281, 708-742. Khirevich, S., & Patzek, T. W. (2018). Behavior of numerical error in pore-scale lattice Boltzmann simulations with simple bounce-back rule: Analysis and highly accurate extrapolation. Physics of Fluids, 30(9), 093604. Zakirov, T., & Galeev, A. (2019). Absolute permeability calculations in micro- computed tomography models of sandstones by Navier-Stokes and lattice Boltzmann equations. International Journal of Heat and Mass Transfer, 129, 415-426.

Thank you for your suggestions. We added a description on how we increased the numerical resolution on page 14 line 13-15. We do not apply any interpolation or stochastic reconstructions to conserve spacial statistics as suggested in the mentioned paper in your comment, but rather used a "magnification" where voxels are subdivided into a certain amount of subvoxels without modifying their phase.

Concerning the convergence from above and below, we added the suggested references and discussed this issue in the corresponding section as well as in the discussion. For the given sample, it is not clear how the method used in Andrä et al. (2013) performs as they only provide results for a single resolution.

10. I would recommend to present a very brief comparison against existing FDM codes, for example FDMSS. I would expect that your code is more accurate, yet takes much longer time to converge and more computationally heavy in terms of CPU and RAM.

It is hard to compare timings of our simulations to other FDM as we used different numerical settings depending on the size of the setup, meaning that the number of cores was varied between simulations with different resolutions (simulations were also partly run on a different cluster). The different computation times are therefore not really comparable. We added an example of the employed number of cores, RAM and timing for one specific simulation (Page 17, line 29-33). Your expectations were quite right, as LaMEM requires more computational resources and also takes more time to converge as e.g. FDMSS., yet LaMEM is therefore more general as it can compute non-Newtonian fluid rheologies.

---

## Author Comment (AC2) · 21 Aug 2019

We thank Stéphane Beaussier for his review. His useful comments helped us to improve our manuscript.

Please find below a point by point response to the comments (comments of the reviewer in black and our response in blue) and the revised version of the manuscript in the supplement.

Sincerely, on behalf of the authors Philipp Eichheimer

1. I could not find any link to this manuscript version of the LaMEM code in the text. If I am correct it is an open source code and therefore it would be necessary to provide the code as an online supplementary or at least a link to the repository in the manuscript.

   We added a link to the open-source repository as well as the revision number, which has been used to reproduce the results of this work. (Page 18, line 4; Page 6, line 1-6)

2. The only issue I have with this manuscript is that I find the information provided on the stencil rescaling a little limited. Given the importance it takes in the manuscript, I would expect a more extended explanation of the method and its implementation in the code. In particular, I believe better discussion on the stability an accuracy of the FD stencil rescaling with references would improve the quality of the manuscript.

   Thank you for your comment. In order to provide more information about stencil rescaling we explained the technique in more detail (Page 6, line 12-25) and modified Figure 1+2 for better understanding. Additionally we also provide more information on stencil rescaling as it has been used in earlier works with application to porous media. We added those as references. These studies performed similar benchmarks e.g. flow between two parallel plates to demonstrate the accuracy of their method. To our knowledge, none of these studies explicitly compared the accuracy of the rescaled stencil approach with the standard staggered grid discretization. As we solve a steady-state problem, we do not have to discretize a time derivative and thus consider the stability of the rescaled stencil method.

3. In page 12 line 2 is written: "using power law exponents ranging from 0.5 to 2." Yet, in figure 6 you only show two values that are tested rather than a range. I would suggest changing the phrasing to "when using 0.5 and 2 as values for the

power law exponents".

We rephrased the sentence as suggested. (Page 13, line 3)

4. In Fig. 3, 4 and 5 black lines with a numerical value are shown but not explained in the caption. It took me quite some time to understand it was the curve local slop. Therefore, I suggest adding a sentence in the caption to explicitly tell what these black lines are, or remove them as the figures are already self-explanatory without giving an numerical value for the local slope. In figure 4 you most likely flipped P3 and P4 in the top left corner schematic as according to line 22-23 page 8 P4 is the largest tube and not P3.

For clarity we added a short sentence in the figure captions. Furthermore as shown in figure 4 tube P3 is the largest tube now referring to the correct value at page 9 line 16-17.

5. In figure 5 the box displaying streamlines around the sphere show significant variations of flow velocity perpendicular to the direction of flow (3 orders of magnitude!!). This is very puzzling as I would expect the flow to be relatively homogenous laterally. Is this a rendering mistake or a consequence of boundary conditions? It should either be corrected or explained in the text.

We changed the figure as the rendered streamlines were not representative and thus the figure was perhaps confusing. Figure 5 now shows computed streamlines of the velocity around the spheres. This should make it easier to understand and highlight the flow structure. (Figure 5)

6. In figure A1: rtol should be added below the x-axis of subfigure a) and b).

We added labels to each x-axis of the corresponding subfigure. (Figure A1)

7. In the authors contribution there is a spelling mistake in the authors initials. OK is written instead of MOK for M. O. Kottwitz.

Sorry, we changed the authors initials. (Page 21, line 3)

**Supplement:**

**Pore-scale permeability prediction for Newtonian and non-Newtonian fluids**

[revised manuscript text omitted]